# Insights into the Microbicidal, Antibiofilm, Antioxidant and Toxicity Profile of New O-Aryl-Carbamoyl-Oxymino-Fluorene Derivatives

**DOI:** 10.3390/ijms24087020

**Published:** 2023-04-10

**Authors:** Ilinca Margareta Vlad, Diana Camelia Nuță, Robert Viorel Ancuceanu, Teodora Costea, Maria Coanda, Marcela Popa, Luminita Gabriela Marutescu, Irina Zarafu, Petre Ionita, Cristina Elena Dinu Pirvu, Coralia Bleotu, Mariana-Carmen Chifiriuc, Carmen Limban

**Affiliations:** 1Department of Pharmaceutical Chemistry, Faculty of Pharmacy, “Carol Davila” University of Medicine and Pharmacy, 020956 Bucharest, Romania; ilinca.vlad@umfcd.ro (I.M.V.); maria.coanda@drd.umfcd.ro (M.C.);; 2Department of Pharmaceutical Botany and Cell Biology, Faculty of Pharmacy, “Carol Davila” University of Medicine and Pharmacy, 6 TraianVuia, 020956 Bucharest, Romania; robert.ancuceanu@umfcd.ro; 3Department of Pharmacognosy, Phytochemistry and Phytotherapy, Faculty of Pharmacy, “Carol Davila” University of Medicine and Pharmacy, 020956 Bucharest, Romania; teodora.costea@umfcd.ro; 4Research Institute of the University of Bucharest—ICUB, University of Bucharest, 50567 Bucharest, Romania; bmarcelica@yahoo.com (M.P.); lumi.marutescu@gmail.com (L.G.M.);; 5Department of Botany & Microbiology, University of Bucharest, 050095 Bucharest, Romania; 6Department of Organic Chemistry, Biochemistry and Catalysis, Faculty of Chemistry, University of Bucharest, 050663 Bucharest, Romania; zarafuirina@yahoo.fr (I.Z.); petre.ionita@unibuc.ro (P.I.); 7Department of Physical and Colloidal Chemistry, Faculty of Pharmacy, “Carol Davila” University of Medicine and Pharmacy, 020956 Bucharest, Romania; 8Ştefan S. Nicolau Institute of Virology, 285 Mihai Bravu Avenue, 030304 Bucharest, Romania; 9Romanian Academy, 050044 Bucharest, Romania

**Keywords:** 9-fluorenone, 9*H*-fluoren-9-one oxime, O-aryl-carbamoyl-oxymino-fluorene derivatives, antimicrobial activity, flow cytometry, *Artemia franciscana* Kellog

## Abstract

The unprecedented increase in microbial resistance rates to all current drugs raises an acute need for the design of more effective antimicrobial strategies. Moreover, the importance of oxidative stress due to chronic inflammation in infections with resistant bacteria represents a key factor for the development of new antibacterial agents with potential antioxidant effects. Thus, the purpose of this study was to bioevaluate new O-aryl-carbamoyl-oxymino-fluorene derivatives for their potential use against infectious diseases. With this aim, their antimicrobial effect was evaluated using quantitative assays (minimum inhibitory/bactericidal/biofilms inhibitory concentrations) (MIC/MBC/MBIC), the obtained values being 0.156–10/0.312–10/0.009–1.25 mg/mL), while some of the involved mechanisms (i.e., membrane depolarization) were investigated by flow cytometry. The antioxidant activity was evaluated by studying the scavenger capacity of DPPH and ABTS•+ radicals and the toxicity was tested in vitro on three cell lines and in vivo on the crustacean *Artemia franciscana* Kellog. The four compounds derived from 9*H*-fluoren-9-one oxime proved to exhibit promising antimicrobial features and particularly, a significant antibiofilm activity. The presence of chlorine induced an electron-withdrawing effect, favoring the anti-*Staphylococcus aureus* and that of the methyl group exhibited a +I effect of enhancing the anti-*Candida albicans* activity. The IC50 values calculated in the two toxicity assays revealed similar values and the potential of these compounds to inhibit the proliferation of tumoral cells. Taken together, all these data demonstrate the potential of the tested compounds to be further used for the development of novel antimicrobial and anticancer agents.

## 1. Introduction

Antibiotics are one of the most important medical discoveries, leading to a remarkable decrease in the mortality and morbidity caused by infectious diseases, but also fostering the progress of modern medicine, by making possible procedures such as transplantation, cancer chemotherapy and surgery [1,2].

Unfortunately, numerous factors, such as the over- and inappropriate use of antimicrobials not only in human medicine but also in the agricultural and veterinary sectors have now led to the occurrence, enrichment and dissemination of antimicrobial resistance (AMR), mirrored in the high prevalence of antibiotic-resistant infections, accompanied by increased mortality rates (25,000 patients are killed each year in Europe by multi-drug resistant bacteria and the estimates of annual deaths will reach 10 million until 2050 if action is not taken) and a huge economic burden [3,4,5,6].

The Centers for Disease Control and Prevention (CDC) has published a list of the 18 most important antibiotic resistance threats classified into three categories depending on the measures required (critical, high, medium) [7]. Additionally, the World Health Organization (WHO) [8] released in 2017 the list of problematic resistant bacteria, grouped under the acronym ESKAPE (*Enterococcus faecium*, *Staphylococcus aureus*, *Klebsiella pneumoniae*, *Acinetobacter baumannii*, *Pseudomonas aeruginosa*, *Enterobacter* sp.), subsequently replaced by ESCAPE (*E. faecium*, *S. aureus*, *Clostridium difficile*, *A. baumannii*, *P. aeruginosa, Enterobacteriaceae*) [9,10,11].

The problem of AMR is amplified by microbial biofilms, which are represented by microbial communities composed of cells adherent to a surface, protected by a matrix of extracellular polymeric substances, expressing a modified phenotype regarding the growth rate and gene transcription, and exhibiting increased tolerance (sometimes hundred up to thousand times higher than their planktonic counterparts) to antibiotics and other chemical inhibitors [12,13].

Antibiotic-resistant strains are currently isolated from hospital- and community-acquired infections and from the natural environment. A so-called “post-antibiotic era” is expected to appear, which would make it impossible to treat infections caused by multidrug-resistant strains. Many international authorities thus advocate identifying incentives to encourage research in the field of antimicrobial drug discovery. 

In this context, the aim of this paper was to evaluate the antimicrobial features of new O-aryl-carbamoyl-oxymino-fluorene derivatives, previously reported for their inhibitory activity against Gram-positive (e.g., *Bacillus anthracis*, *S. aureus*, including methicillin resistant strains) and Gram-negative (*Escherichia coli*, *Proteus mirabilis*, *K. pneumoniae*, *P. aeruginosa*, *Burkholderiathailandensis* and *Francisella tularensis*) bacterial strains, mycobacteria, yeasts and molds [14,15,16,17]. The antioxidant activity of these compounds has been also evaluated, as it could represent an advantage for novel antimicrobial leads by decreasing the intensity and duration of the inflammatory response often accompanying the infectious process, thus avoiding their deleterious effects on the host tissues [18]. It is well known that oxidative stress represents an imbalance between the generation of free radicals and a decrease in the concentration of endogenous antioxidants (such as glutathione, vitamin C, vitamin E and a series of enzymes—catalase, superoxide dismutase and peroxidases) [19]. Free radicals (ROS—reactive oxygen species or RNS—reactive nitrogen species) contain more than one unpaired electron, which is unstable and attacks proteins, nucleic acids and lipids [19]. Generally, ROS include superoxide anion (O_2_^−^), hydrogen peroxide (H_2_O_2_), hydroxyl radical (OH∙), singlet oxygen or nitric oxide (NO) [19]. Endogenous sources of free radicals include the mitochondrial respiratory chain, mental stress, aging, inflammation, or ischemia/reperfusion [20,21]. Free radicals have a dual behavior, as at higher concentrations they have shown negative effects upon the biological system (being involved in autoimmune, cardiovascular, neurodegenerative, and metabolic diseases or cancer) [22], whilst low/moderate amounts have beneficial properties (modulation of different signaling pathways, phagocytosis, mitogenic response, etc.) [22]. However, infections with resistant bacteria (*Staphylococcus aureus*, *Escherichia coli* or *Pseudomonas aeruginosa*, *Proteus* sp.) [23,24,25,26], as previously mentioned, lead to chronic inflammation with the continuous generation of cytokines and chemokines by macrophages (such as interleukins IL-4, IL-5, IL-12 or tumor necrosis factor TNF-α). The activation of macrophages is further involved in the modulation of several pathways (mediated by nuclear factor kappa B, activator protein 1, nuclear factor of activated T cells, hypoxia-inducible factor 1-α), with the ongoing hyperproduction of ROS and RNS. An important consequence of chronic inflammation is tissue damage due to persistent oxidative stress and the excessive induction of tissue repair mechanisms [27].

Fluorene is a polycyclic aromatic hydrocarbon consisting of two benzene rings joined by a direct carbon-carbon bond and an adjacent methylene bridge. The fluorene nucleus is found in numerous bioactive molecules. Different derivatives are studied or even used for the treatment of infectious, metabolic, cardiovascular, neoplastic, immunological and neuromuscular diseases. For example, the fluorenic nucleus is present in the structure of lumefantrine, with antimalarial properties, hexafluronium bromide, a muscle relaxant acts by inhibiting the cholinesterase [28], pavatrine is a spasmolytic agent [29,30], indecainide is an antiarrhythmic agent of class Ic [31], alconyl and its derivatives with aldose reductase inhibitory activity are used in treating diabetogenic cataract and neuropathy [32,33,34,35,36,37], cycloprofen and leumedins have anti-inflammatory activity [38,39,40,41]. Some other derivatives are investigated for their hypoglycemic activity and for decreasing insulin resistance [42], for their antiviral properties, including against human immunodeficiency virus (HIV-1) and severe acute respiratory syndrome coronavirus 2 (SARS-CoV-2) virus [43,44,45,46,47,48,49,50,51,52], anticoagulant effect [53], antitumoral activity [54], immunosuppressive [55] or cardiodepressant activity [56]. The fluorenic fragment is also found in many natural products.

## 2. Results

### 2.1. Antimicrobial Activity against Planktonic and Biofilm Embedded Microbial Cells

The tested compounds showed an inhibitory effect on microbial growth at minimum inhibitory concentration (MIC) values of 0.156–10 mg/mL (Table 1) and minimum bactericidal concentration (MBC) values between 0.312 and 10 mg/mL (Table 2). The four compounds inhibited the ability of bacterial and fungal strains to adhere and develop biofilms on the inert substratum, with a minimum biofilm inhibitory concentration (MBIC) of 0.009–1.25 mg/mL (Table 3). 

### 2.2. Elucidation of the Potential Mechanisms of Antimicrobial Action by Flow Cytometry (FCM)

The FCM assay results have shown that the green fluorescence of the potential-sensitive probe, DiBAC4(3), was enhanced at subinhibitory concentrations, supporting the hypothesis that the in vitro bactericidal activity of the tested compounds was the result of cytoplasmic membrane potential dissipation. However, it has been suggested that membrane depolarization is required to facilitate the entry of antibiotics into bacteria for the expression of activity. Additionally, the disruption of membrane function may actually have intracellular targets [57]. Thus, further studies are needed for the elucidation of the mechanisms of antimicrobial action.

The compound **1c** did not cause changes in the membrane potential of the two Gram-positive strains, i.e., *E. faecalis* ATCC 29212 and *S. aureus* ATCC 25923. Table 4 shows the membrane depolarization demonstrated by the increased values of the staining index (SI).

### 2.3. The Toxicity Profile of the Tested Compounds on the Artemia franciscana Kellog Model

Three of the four compounds (**1a**–**c**) were not toxic at the tested concentrations, all nauplii were alive and showing normal movements. Compound **1d** manifested moderate toxicity, as evidenced by the lethality curve (concentration-response) (Figure 1) and by the LC50 value (14.63 μg/mL, 95% CI 11.80–18.15 μg/mL).

### 2.4. Antioxidant Activity Evaluated by Scavenger Activity towards DPPH and ABTS^•+^ Free Radicals

The antioxidant activity was tested only for the compounds **1a**–**c** (tested in a two-fold concentration range from 25 to 1000 μM), which proved no cytotoxicity in the previous *Artemia franciscana* Kellog in vivo assay. The absorbance values for all analyzed compounds decreased with the increase in concentration. varying between 0.9310 nm (at 25 μM) and 0.8547 nm (at 1000 μM) for compound **1a**; 0.9269 nm (at 25 μM) and 0.8923 nm (at 1000 μM) for compound **1b** and 0.8770 nm (at 25 μM) and 0.8026 nm (at 1000 μM) for compound **1c**. The results of the DPPH free radical scavenger activity are presented in Figure 2. DPPH free radical scavenger capacity varies between 12.66% (**1a** at the concentration of 25 μM) and 24.64 % (**1c** at the concentration of 1000 μM). Moreover, compound **1a** inhibited with 19.81% free radical activity at the maximum concentration of 1000 μM. The lowest scavenger activity recorded at the highest tested concentration of 1000 µM was observed for compound **1b** (16.29%). However, since the scavenging activity was low (below 30%), all tested compounds have shown very high IC50 (μM) values beyond 1000 μM (Table 5), which is a strong indicator of the low antioxidant potential. Still, the lowest IC50 value, which indicates the best antioxidant activity, was obtained for compound **1c**, followed by compounds **1a** and **1b** (Table 5). Significant differences have been found between IC50 values for all tested compounds (*p* < 0.0001) (Table 6).

In the second assay, the absorbance of ABTS^•+^ free radical solution in the presence of tested compounds varied between 0.5802 (at 25 μM) and 0.5564 (at 1000 μM) for compound **1a**, 0.5332 (at 25 μM) and 0.5109 (at 1000 μM) for compound **1b**, 0.5848 (at 25 μM) and 0.5658 (at 1000 μM) for compound **1c**. The scavenger activity increases with concentration, regardless of the analyzed compound (Figure 3). Free radical scavenger (Table 7) activity varied between 12.53% (**1c** at 25 μM) and 26.60% (**1b** at 1000 μM). For the tested concentration range, the highest scavenger activity was observed for compound **1b**, and the lowest for compound **1c**. In comparison with the DPPH assay, in the ABTS assay, a higher scavenging activity was observed for **1a**, and **b** at all tested concentrations, while for **1c**, the inhibition was lower.

All tested compounds exhibited a low antioxidant effect, with high IC50 (μM) values beyond 1000 μM (Table 7). Analyzing Table 7, one can note that the best antioxidant activity was observed for 1b followed by **1a,c**. These differences among antioxidant assays are probably the consequence of the compound lipophilicity and specific mechanism of action against free radicals. On the other hand, the higher antioxidant effect of compound **1b** can be explained by the inductive effect of the methyl group that confers greater molecular stability. Significant differences have been found between IC50 values for all tested compounds (*p* < 0.0001) (Table 8).

### 2.5. Cytotoxicity

The cytotoxicity of the investigated compounds was tested using the IncuCyte Basic Analysiskit, allowing to calculate the IC50, against three tumoral cell lines, i.e., HeLa (cervical cancer cells), HT29 (colon adenocarcinoma) and MG63 (osteosarcoma). The selected metric of the concentration-response module was set on phase and confluence (%). The IC50 levels are presented in Table 9. The new derivatives exhibited cytotoxicity at concentrations lower than 100 µg/mL, from 6.33 ± 3.02 to 31.5 µg/mL, as observed from the IC50 calculation, the susceptibility of the tested cell lines increasing from MG63 cells to HeLa cells.

## 3. Discussion

Less than half a century after the discovery of antibiotics, we are now threatened to enter the post-antibiotic era, where the fatality rate due to infections will increase sharply, particularly in less developed countries and in the infant population. A lot of modern medical procedures, such as transplants, surgery or chemotherapy will no longer be possible because of infections with multidrug-resistant bacteria. Thus, there is an acute need to find antibiotics with original structures, having new microbial targets. Taking into account that fluorine is found in diverse pharmacologically active compounds with antimalarial (lumefantrine), antiarrhythmic (indecainide), muscle relaxant (hexafluronium bromide or antiviral (tilorone) activity as well as the pharmacological activities of carbamoyl and oximinic pharmacophore groups, we have combined these biologically active fragments into a single original molecule, to obtain new compounds of the class O-aryl-carbamoyl-oxymino-fluorene, previously characterized [17] (Figure 4) and to evaluate their potential bactericidal, fungicidal and antibiofilm effects.

The most susceptible strains in planktonic growth were *S. aureus*, followed by the *P. aeruginosa* strain. The most active compound against planktonic cells was **1d**, which exhibited the lowest MIC value of 0.156 mg/mL against the *S. aureus* strain. The MBC values were similar or twice as high as the MIC ones, indicating that bactericidal activity and membrane depolarization were correlated for compound **1d**-treated *S. aureus* cells, suggesting a dose-dependent bactericidal effect of the tested compounds on the membrane integrity, as revealed by the FCM analysis.

The biofilm formed by *E. faecalis* manifested an increased susceptibility to compound **1c**, *S. aureus* to **1d**, *P. aeruginosa* to **1a**, *E. coli* to **1b** and that of *C. albicans* to all tested compounds, and mainly to **1b**. The MBEC values have been significantly lower (up to hundreds of times) than the corresponding MIC and MBC ones.

Taken together, the results of the antimicrobial activity assays suggest that the electron-withdrawing inductive effect of chlorine atoms enhanced the activity against planktonic and adhered *S. aureus*, while the +I effect of the methyl group enhanced the anti-fungal activity against *C. albicans* strain.

The evaluation of the undesired cytotoxic effects of new molecules aimed to be developed as a pharmaceutical is crucial to ensure drug safety and effectiveness. The toxicity on the *Artemia franciscana* Kellog crustacean species was assessed based on the method of B. N. Meyer et al. [58] and T.W. Sam [59], with slight adaptations suggested by more recent sources [60,61,62]. Robust methods were used to model the concentration-response relationship and to calculate the IC50 and IC 95% values. It must be underlined that an IC50 value of 10–30 μg/mL corresponds to moderate toxicity, as revealed in the case of compound **1d** exhibiting an acute toxicity of 14.63 μg/ mL, which is very close to the one of cyclophosphamide, which has an IC50 value of 16.3 μg/mL [63,64]. The other evaluated compounds (**1a**, **1b** and **1c**) were not toxic at concentrations up to 100 μg/mL, therefore, at the solubility limit. The IC50 calculated for the four compounds using the in vitro cytotoxicity assay on three tumoral cell lines ranged between 6 and 32 μg/mL. These findings provide insights into the further investigation of these derivatives for their potential as a therapeutic targeting rapidly dividing cancer cells. Furthermore, in our future research, we will establish the cellular targets and pathways activated by these derivatives, which will provide insights into their mechanism of action.

The antioxidant activity was tested by two assays, i.e., the scavenger activity towards the DPPH and ABTS^•+^ free radicals. The DPPH (2,2-diphenyl-1-picrylhydrazyl) is a colored free radical that is reduced by antioxidants to pale-yellow hydrazine. The color change leads to a reduction in absorbance values [65,66,67]. Briefly, the method is based on electron transfer, although some authors consider that both electron and hydrogen atom transfer processes are involved. The method’s advantages are that it is simple, and inexpensive and the provided results are well correlated with those obtained by other methods [3,68,69]. The ABTS^•+^ free radical is produced in the reaction of 2,2′-azinobis-(3-ethylbenzothiazoline-6-sulfonic acid) diammonium salt with manganese (II) oxide [70], sodium/potassium persulfate [71], 2,2’-azobis(2-amidinopropane) dihydrochloride [72] or enzymes (peroxidase) [6]. The free radical has a blue color, with the ABTS^•+^ free radical being reduced in the presence of oxidants; the blue color loss is accompanied by a decrease in absorbance [3,67]. The method has several advantages: it is simple and reproducible, it does not depend on pH and can be used for the evaluation of both lipophilic and hydrophilic compounds. The method is based mainly on proton transfer [6]. The reaction time between ABTS and the substrate varies between 4–6 and 60 min [71,73].

Generally, the scavenger capacity towards both free radicals for the tested concentrations was below 30%. Significant differences have been found between IC50 (μM) values) of tested compounds by means of both antioxidant assays. The differences among DPPH and ABTS assays, between the analyzed compounds, are probably due to the lipophilicity of the compounds or to the specific mechanism of the antioxidant agent. Although the antioxidant activity is desirable for protecting the host cells from the toxic effects of an antibiotic, there are studies raising awareness that at least in anaerobic environments, the antioxidant activity could interfere with the ROS-mediated lethality of bactericidal antibiotics such as ampicillin, gentamicin or norfloxacin. The pretreatment with glutathione and ascorbic acid antioxidants decreased the lethality induced by ampicillin, gentamicin, and norfloxacin by at least 1-log at 4 h posttreatment [74].

## 4. Materials and Methods

### 4.1. Tested Compounds

We have previously synthesized and characterized four new O-aryl-carbamoyl-oxymino-fluorene derivatives (**1a**–**d**) (Figure 1) [17].

### 4.2. Microbiological Assays

The inhibitory activity of the obtained compounds has been assessed on five microbial strains, respectively: *E. coli* ATCC 25922, *P. aeruginosa* ATCC 27853, *S. aureus* ATCC 25923, *E. faecalis* ATCC 29212 and *C. albicans* ATCC 90029, in a planktonic and adherent growth state.

The MIC assay was performed by the serial two-fold microdilutions in a liquid medium, using a concentration range from 5 to 0.009 mg/mL and microbial suspensions of 0.5 MacFarland density. The wells containing only the microbial culture served as positive control and those containing the sterile culture medium as the negative control. The MIC values were read after incubation at 37 °C for 24 h [75].

The MB assay was performed after reading the MIC values. For this purpose, 10 µL volumes of the liquid culture developed in the wells containing the highest concentration to the MIC value were plated on a solid culture medium to determine the MBC value (the concentration that has totally inhibited the microbial growth).

The anti-biofilm activity assay was performed by the purple violet microtiter method, as previously described, allowing us to determine the minimal biofilm inhibitory concentration (MBEC). The same range of concentrations, as in the MIC assay, i.e., from 5 to 0.009 mg/mL has been tested [76].

The investigation of potential mechanisms of antimicrobial action by flow cytometry (FCM) was performed on microbial cultures obtained after 18–24 h of microbial cell incubation in the presence of subinhibitory concentrations of the analyzed compounds. Thus, microbial suspensions with a density of approximately 10^6^ CFU/mL were prepared in sterile saline phosphate buffer from exponentially growing microbial cultures, obtained on a solid culture medium. Subinhibitory concentrations of the compounds were prepared in Muller–Hinton liquid culture medium and inoculated with an equal amount of microbial suspension and incubated for 18–24 h at 37 °C.

Microbial suspensions inoculated in a liquid medium were used as growth control. After incubation, the DiBAC4 fluorochrome solution [bis-(1,3-dibutylbarbituric acid) trimethinoxonol] Invitrogen/Life Technologies, Carlsbad, was added (0.5 μg/mL).

Fluorescence intensity (FI) was measured with an Accuri C6 plus flow cytometer in the FITC fluorescence channel. Growth control was used to locate the microbial cell population for fluorescence measurements. DiBAC4 dye was used to detect changes in the microbial membrane potential. The fraction of microbial cells in the analyzed population that showed increased green fluorescence (corresponding to membrane depolarization) was calculated after the exclusion of untreated growth control fluorescence. A twofold increased fluorescence intensity (median fluorescence intensity = MFI) was considered to correspond to the depolarized microbial cells. For each subinhibitory concentration, a coloring index (CI) was calculated which represents the ratio of fluorescence intensity (FI) of treated versus untreated cells [77].

### 4.3. The Toxicity on the Artemia Franciscana Kellog Crustacean Species

The oocysts (Great Salt Lake, USA) provided by S.K. Trading were grown on artificial seawater medium (Coral Marine, Grotech), dissolved in distilled water with a few minutes of sonication, at 30 g/L concentration. The test was carried out in a 24-well plate, using three replicates. Because of their low solubility, the compounds **1a**–**d** were suspended in the culture medium using sodium alginate (0.045%) (also used a negative control), the testing being carried out at the solubility limit. The concentrations 100, 50, 25, 12.5 and 6.2 μg/mL used for each substance were obtained by successive dilutions starting from the initial suspension. Between 10 and 15 nauplii per well were collected and placed in contact with the test suspensions (1.5 mL/well). All nauplii, dead or alive, were counted and recorded within 24 h of being placed in contact with the tested suspensions. The non-linear modeling of the concentration-lethality relationship was achieved through a four-parameter logistics model (4PL), implemented in several robust variants for estimating parameters in the R package “dr4pl” [78].

### 4.4. In Vitro Cytotoxicity Assay

In our experiments, we used colorectal adenocarcinoma HT29 (ATCC HTB-38), cervical adenocarcinoma cells HeLa (ATCC CRM-CCL2), and osteosarcoma MG63 (ATCC CRL-1427). The cells were maintained in Dulbeco’s Modified Medium (DMEM): F12 (Thermo Fisher Scientific, Waltham, MA, USA) supplemented with 10% fetal bovine serum (FBS, Thermo Fisher Scientific, Waltham, MA, USA). For the in vitro evaluation of their cytotoxicity, the new derivatives were solubilized in DMSO at a concentration of 10 mg/mL and filtered with 0.22 µm Millex-GV Filter (Merck, Darmstadt, Germany). For cytotoxicity kinetics, the IncuCyte^®^ S3 Live-Cell Analysis System (Sartorius, AG, Goettingen, Germany) was used. Briefly, 2 × 10^4^ cells seeded in 96 well plates were treated with the new derivatives in a concentration ranging between 500 μg/mL and 3.9 μg/mL. The plates were placed in the IncuCyte^®^ S3 Live-Cell Analysis System, and five images per well were taken every six h over a 72-h period and then processed, according to kit recommendations.

### 4.5. Antioxidant Activity of Compounds

Scavenger Activity towards DPPH Free Radical

The DPPH (Sigma-Aldrich, Darmstadt, Germany) solution of 0.1 mM concentration was obtained by dissolving 0.0039 g of free radical in 100 mL ethanol, in a volumetric flask. For all determinations, the DPPH solution was freshly prepared and kept in the dark. Ethanol has been chosen since tested compounds were dissolved in ethanol: DMSO = 99:1 (*v*/*v*) mixture.

The analyzed compounds were dissolved in 25 mL of ethanol: DMSO = 99:1 (*v*/*v*) mixture in a final concentration of 1000 μM (stock solution), from which several concentrations of 25 μM, 50 μM, 75 μM, 100 μM, 250 μM and 500 μM were obtained by successive dilutions.

The determination of the antioxidant activity was based on the Ohnishi M et al. method [79], quoted by Germano M.P. et al. [80].

For this purpose, 0.5 mL of 25–1000 μM tested solutions were treated with 3 mL DPPH solution (0.1 mM). The samples were kept at rest in the dark for 30 min [79,81]. Ethanol was used as a blank in order to measure the absorbance at 515 nm (Jasco V-530 spectrophotometer, Jasco, Japan).

The following formula was used to determine DPPH free radical scavenger activity (%) [5,82]:$$Inhibition\;\%=\frac{A_{control}-A_{sample}}{A_{control}}\times 100$$
where, *A* = absorbance of the 0.1 mM DPPH solution in the absence (*control*)/*presence* (*sample*) of tested compounds after 30 min.

Scavenger Activity towards ABTS^•+^ Free Radical

The method according to Re R. et al. was used to evaluate the scavenger capacity of the free radical [71].

The ABTS^•+^ free radical resulted from the reaction between 2,2′-azinobis-(3-ethylbenzothiazoline-6-sulfonic acid) diammonium salt (7.4 mM) (solution 1) (Sigma-Aldrich, Germany) and potassium persulfate (2.6 mM) (solution 2) (Roth, Germany) [70].

In order to obtain the ABTS^•+^ free radical reagent, equal volumes of solutions 1 and 2 were mixed for 16 h. The obtained reagent was kept in the dark; 1 mL of the obtained solution was brought to a 50 mL volumetric flask and diluted with ethanol so that the absorbance at λ = 734 nm would be 0.700 ± 0.02.

The tested compounds were dissolved and diluted similar to the previous antioxidant assay.

A volume of 0.5 mL of 25–1000 µM solutions was treated with 3 mL ABTS^•+^ solution, stirred and kept in the dark for 6 min and then, the absorbance was read at 734 nm (Jasco V-530 spectrophotometer) using ethanol as a blank.

The following formula was used for the calculation of the ABTS^•+^ free radical scavenger activity (%)
$$Inhibition\;\%=\frac{{Abs}_{\left(t=0min\right)}-{Abs}_{\left(t=6min\right)}}{{Abs}_{\left(t=0min\right)}}\times 100$$
where: *Abst* = *ABTS*^•+^ solution absorbance in the absence (0 min)/presence (6 min) of the tested compounds.

For both the above-mentioned methods, the antioxidant activity was expressed as IC_50_ values (μM) which represent the concentration of each compound for which the scavenging activity of free radicals is 50%. The IC_50_ values were calculated by linear regression plots, where the abscissa was represented by the concentration of the tested compound solution (25–1000 μM) and the ordinate the average percent of antioxidant capacity from three separate tests.

### 4.6. Statistical Analyses

For each tested concentration, all the determinations were carried out in triplicate; we established the mean ± standard deviation (SD) of three independent determinations. Microsoft Office (Excel 2007) and GraphPad Prism v.5 (GraphPad, SUA) were used to perform the statistical analysis. The antioxidant activity of the analyzed compounds was compared using the one-way ANOVA test followed by the Tukey post-test (*p* < 0.05 for statistical significance).

## 5. Conclusions

A series of four O-aryl-carbamoyl-oxymino-fluorene derivatives previously synthesized using the intermediate compound 9*H*-fluoren-9-one oxime and previously characterized have been bioevaluated in this paper, to formulate potential leads for their biomedical applications.

The four derivatives proved to inhibit the tested microbial strains’ growth, in both planktonic and adherent states. The electron-withdrawing inductive effect of chlorine atoms enhanced the anti-staphylococcal activity, both against free-floating and adherent cells, while the +I effect of the methyl group favored the anti-fungal activity. Thus, they can be considered for further antimicrobial agent development.

The analysis of the effects on the membrane potential of the tested microbial strains showed that, at subinhibitory concentrations, they produce a depolarization of the plasma membrane, which is thus one of the targets of their antimicrobial activity.

The compounds were evaluated for their in vitro cytotoxicity on three cell lines and in vivo acute toxicity. All four tested compounds exhibited a similar profile of cytotoxicity on the three cellular lines, HeLa, HT29 and MG63, while three of them (**1a**–**c**) were non-toxic at the solubility limit on *Artemia franciscana* Kellog model.

The compounds have shown a modest scavenger capacity towards the DPPH free radical (<30%) in the tested concentration range, the most active being **1a**,**c**. All tested compounds have shown scavenger activity towards the ABTS•+ free radical in the tested concentration range, with compound **1b** exhibiting the best antioxidant activity, correlated with the +I effect and electron donating tendency of the methyl group. The antioxidant activity of these compounds could represent an advantage for novel antimicrobial leads, acting by decreasing the intensity and duration of the inflammatory response often accompanying the infectious process, thus avoiding their deleterious effects on the host tissues.

Taken together, all these data demonstrate the potential of the tested compounds to be further used for the development of novel antimicrobial and anticancer agents.

## Figures and Tables

**Figure 1 ijms-24-07020-f001:**
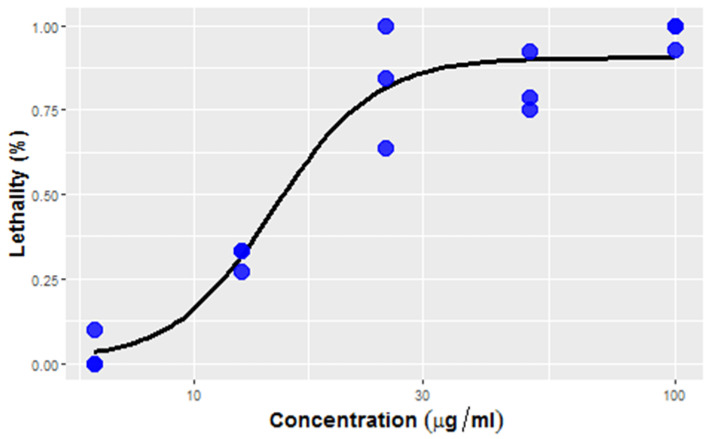
Concentration-response curve for the lethality of substance**1d** on *Artemia franciscana* Kellog nauplii, built based on a logistic model with four parameters (the *x*-axis corresponds to a logarithmic scale. The blue dots represent the point estimates of lethality for each of the three replicas.

**Figure 2 ijms-24-07020-f002:**
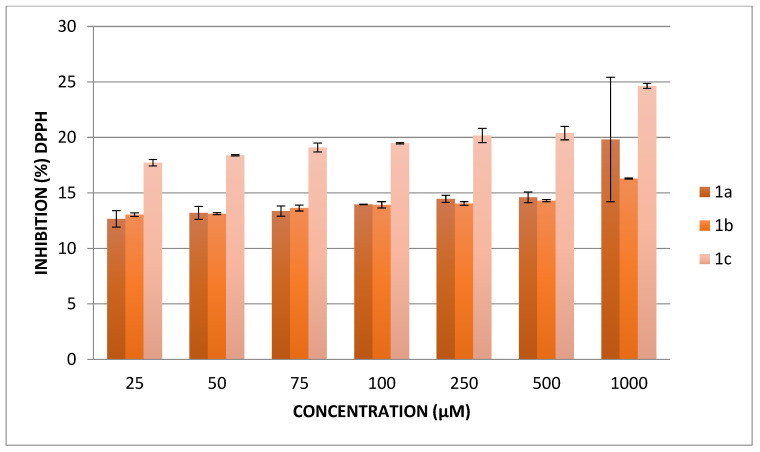
Graphic representation of DPPH free radical inhibition by the analyzed compounds (columns correspond to compounds **1a**–**c** from left to right).

**Figure 3 ijms-24-07020-f003:**
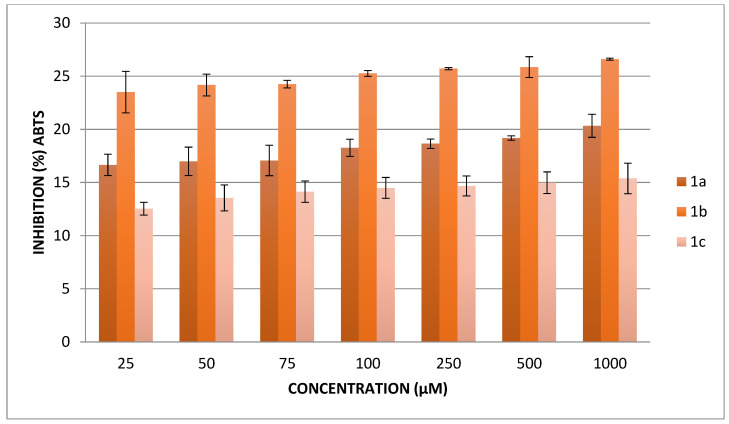
Graphic representation of ABTS^•+^ free radical inhibition by the analyzed compounds (columns correspond to compounds **1a**–**c** from left to right).

**Figure 4 ijms-24-07020-f004:**
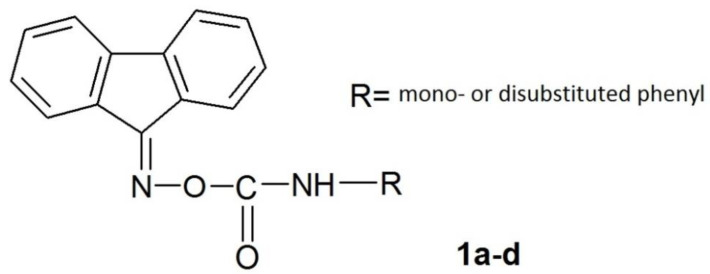
The chemical structure of the new O-aryl-carbamoyl-oxymino-fluorene derivatives (**1a**–**d**). **1a** (R = -C_6_H_5_): 9-(phenylcarbamoyloxymino)fluoren. **1b** (R = -C_6_H_4_(CH_3_)(3)): 9-((3-methyl-phenyl)carbamoyloximino)fluoren. **1c** (R = -C_6_H_4_(Cl)(3)): 9-((3-chloro-phenyl)carbamoyloxymino)fluoren. **1d** (R = -C_6_H_4_(Cl)_2_(3,4)): 9-((3,4-dichloro-phenyl)carbamoyloxymino)fluoren.

**Table 1 ijms-24-07020-t001:** The minimum inhibitory concentration (MIC) values (mg/mL) of the tested compounds.

Microbial Strain	Chemical Compound			
	1a	1b	1c	1d
*E. faecalis* ATCC 29212	2.5	10	5	5
*S. aureus* ATCC 25923	2.5	2.5	2.5	0.156
*P. aeruginosa* ATCC 27853	2.5	2.5	2.5	2.5
*E. coli* ATCC 25922	2.5	5	2.5	5
*C. albicans* ATCC 90029	2.5	5	5	5

**Table 2 ijms-24-07020-t002:** The minimum bactericidal concentration (MBC) values (mg/mL) of the tested compounds.

Microbial Strain	Chemical Compound			
	1a	1b	1c	1d
*E. faecalis* ATCC 29212	5	10	5	5
*S. aureus* ATCC 25923	5	5	5	0.312
*P. aeruginosa* ATCC 27853	2.5	2.5	2.5	2.5
*E. coli* ATCC 25922	2.5	5	5	5
*C. albicans* ATCC 90029	5	10	10	10

**Table 3 ijms-24-07020-t003:** The minimum biofilm inhibitory concentration (MBIC) values (mg/mL) of the tested compounds.

Microbial Strain	Chemical Compound			
	1a	1b	1c	1d
*E. faecalis* ATCC 29212	1.25	1.25	0.312	1.25
*S. aureus* ATCC 25923	5	2.5	5	0.019
*P. aeruginosa* ATCC 27853	0.009	0.156	1.25	1.25
*E. coli* ATCC 25922	0.625	0.078	1.25	0.625
*C. albicans* ATCC 90029	0.312	0.078	0.312	0.312

**Table 4 ijms-24-07020-t004:** The staining index (SI) (mg/mL) values obtained for the tested compounds at subinhibitory concentrations.

Microbial Strain	Chemical Compound	1a	1b	1c	1d
*E. faecalis* ATCC 29212	Subinhibitory concentrations tested	2.5	2.5	1.25	1.25
	staining index value	2.21	3.99	0.32	44.54
*S. aureus* ATCC 25923	Subinhibitory concentrations tested	2.5	2.5	2.5	1.25
	staining index value	3.5	8.17	0.19	4.2
*P. aeruginosa* ATCC 27892	Subinhibitory concentrations tested	2.5	2.5	2.5	2.5
	staining index value	6.68	6.5	2.38	20.45
*E. coli* ATCC 25922	Subinhibitory concentrations tested	1.25	2.5	1.25	1.25
	staining index value	80.7	10.2	3.2	173.3
*C. albicans* ATCC 90029	Subinhibitory concentrations tested	2.5	2.5	-	2.5
	staining index value	4.94	2.82	-	20.21

**Table 5 ijms-24-07020-t005:** IC50 (μM) values for analyzed compounds by means of DPPH assay.

Compound	IC50 (μM)
**1a**	5735.23 ± 0.0828
**1b**	12262.66 ± 0.0574
**1c**	5208.03 ± 0.1245

**Table 6 ijms-24-07020-t006:** Statistical analysis of the antioxidant activity of analyzed compounds by means of DPPH assay.

Tukey’s Multiple Comparisons Test between IC50 Values	Mean Difference	95% CI of Difference	*p* Value
Compound **1a** vs. compound **1b**	−6527	−6528 to −6527	<0.0001 (***)
Compound **1a** vs. compound **1c**	527.2	527.1 to 527.3	<0.0001 (***)
Compound **1b** vs. compound **1c**	7055	7055 to 7055	<0.0001 (***)

CI—confidence interval; *** = *p* < 0.001.

**Table 7 ijms-24-07020-t007:** IC50 (μM) values for analyzed compounds by means of ABTS^•+^ assay.

Compound	IC50 (μM)
**1a**	9885.38 ± 0.2514
**1b**	9379.42 ± 1.0247
**1c**	18,165.5 ± 0.5478

**Table 8 ijms-24-07020-t008:** Statistical analysis of the antioxidant activity of analyzed compounds by means of ABTS^•+^ free radical assay.

Tukey’s Multiple Comparisons Test between IC50 Values	Mean Difference	95% CI of Difference	*p* Value
Compound **1a** vs. compound **1b**	−506	−506.8 to −505.2	<0.0001 (***)
Compound **1a** vs. compound **1c**	−8786	−8787 to −8785	<0.0001 (***)
Compound **1b** vs. compound **1c**	−8280	−8281 to −8279	<0.0001 (***)

CI—confidence interval; *** = *p* < 0.001.

**Table 9 ijms-24-07020-t009:** The IC50 (µg/mL) levels of the tested derivatives.

	1a	1b	1c	1d
HeLa	7.52 ± 1.39	8.5 ± 2.26	8.9 ± 2.56	7.59 ± 3.64
HT29	11 ± 0.75	6.33 ± 3.02	10.8 ± 1.36	8.17 ± 2.98
MG63	31.5	22 ± 2.48	22.8 ± 0.78	26.3 ± 0.92

## Data Availability

Not applicable.

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
