# Peer review of "Insights into the Microbicidal, Antibiofilm, Antioxidant and Toxicity Profile of New O-Aryl-Carbamoyl-Oxymino-Fluorene Derivatives"

_ijms, 2023, doi:10.3390/ijms24087020_

Round 1
Reviewer 1 Report
The submitted manuscript describes the antimicrobial and antioxidant effects of of New O-aryl-carbamoyl-oxymino-fluorene Derivatives. Although the results are interesting, the authors needs to improve the work as in the current form it is not suitable for publication in a high standard journal such IJMS.
Minor issues
- The abstract should be more informative by providing the results, such as MIC and MBC values.
- Please attempt to the use of italics for Latin words such in vitro.
- Please provide the meaning for each abbreviation at the first citation. For instance in the lines 97-100.
- The captions of tables should be improved. Avoid the use of abbreviation.
- The tables are not formatted following the journal standard. Please see the journal template.
Major concerns
- Despite the title, insights into the action mechanisms of the compounds are not properly provided. In fact, only one assay using flow cytometry was performed.
- The importance of antioxidant properties was not addressed in the introduction section.
- Antimicrobial evaluation should be better elucidate. For instance, time-kill curves should be performed to better characterize the antimicrobial effects of the drugs.
- Since the authors address the drug resistance as a major problem, they should evaluate the effects of their compounds against drug resistant strains.
- The antibiofilm assay should be better described, it is not clear how many concentrations were evaluated and for how long the biofilm was formed prior the incubation with the compounds.
- Importantly, it is not clear if the authors performed a biofilm inhibition assay or an eradication assay.
- Please provide the IC50 for each compound in DPPH and ABTS assays.
- The toxicity should be analyzed also using mammalian cells.
Author Response
We are very grateful to the reviewer for the valuable suggestions. We have carefully addressed them in the revised version of the manuscript and briefly offer below our answers point by point to the reviewer's comments.
Minor issues
- The abstract should be more informative by providing the results, such as MIC and MBC values.
We have modified the abstract according to reviewer’s suggestions.
- Please attempt to the use of italics for Latin words such in vitro.
We have used the Latin characters.
- Please provide the meaning for each abbreviation at the first citation. For instance in the lines 97-100.
We have checked and explained the abbreviation when used for the first time.
- The captions of tables should be improved. Avoid the use of abbreviation.
We have improved the tables captions.
- The tables are not formatted following the journal standard. Please see the journal template.
We have formatted the tables according to journal standard.
Major concerns
- Despite the title, insights into the action mechanisms of the compounds are not properly provided. In fact, only one assay using flow cytometry was performed.
We are aware that only some of the mechanisms of action have been revealed. We have stated in the Discussion section that further research is needed to elucidate the mechanisms of action of these compounds.
- The importance of antioxidant properties was not addressed in the introduction section.
We thank the reviewer for the observation. The introduction was completed with data related to the antioxidant activity.
Antimicrobial evaluation should be better elucidate. For instance, time-kill curves should be performed to better characterize the antimicrobial effects of the drugs.
- Since the authors address the drug resistance as a major problem, they should evaluate the effects of their compounds against drug resistant strains.
We are very grateful for these suggestion. Unfortunately, this will take time and resources which were not previewed for this stage of research.
- The antibiofilm assay should be better described, it is not clear how many concentrations were evaluated and for how long the biofilm was formed prior the incubation with the compounds.
We have added more details in the Material and methods section.
- Importantly, it is not clear if the authors performed a biofilm inhibition assay or an eradication assay\.
We have performed a biofilm inhibition assay. This has been tested in the Material and methods section now.
- Please provide the IC50 for each compound in DPPH and ABTS assays.
We have added IC50 (μM) values for both antioxidant assays – see tables 5 and 7 in the manuscript. The IC50 values are very high, since the scavenger activity of all tested compounds against DPPH and ABTS free radicals was low (below 30%), independent of concentration or antioxidant assays.
- The toxicity should be analyzed also using mammalian cells.
We have added the in vitro cytotoxicity on three cellular lines (HeLa, H29 and MG63).
Reviewer 2 Report
In this work, Vlad and colleagues have developed the study and bioevaluation of a series of 4 O-aryl-carbamoyl-oxymino-fluorene derivatives, exploring their antimicrobial, antioxidant and toxicity attributes.
The authors fully describe the set of results obtained and discuss the major conclusions; however, they present a case for the plasma membrane to be one of the targets of the mode of action, substantiated solely on a membrane depolarization assay. This statement would require some further mechanistic discussion, since the result observed with the DiBAC4 fluorochrome could be the result of non-specific membrane damage, considering the lipophilicity of the compounds under study, as well as a downstream effect of a metabolic inhibition cascade leading to loss of cellular fitness and/or inhibition of energy production. Hence, the reported mechanistic conclusions should be presented as one of many possibilities, or further substantiated.
Furthermore, no relevant discussion was present concerning the solubility limits for the compounds tested, reported as 100 μg /mL for distilled water, and, if applicable, the potential impact in the antimicrobial studies, since the concentrations tested were up to 50 times this concentration, even if in diverse media.
Also, the manuscript could be further improved in some minor points:
- No clear designation of the quality control antibiotics for antimicrobial and anti-biofilm assays.
- Data presented in tables should be standardized in respect to decimal places and/or significant digits
Author Response
We are very grateful to the reviewer for the valuable suggestions. We have carefully addressed them in the revised version of the manuscript and briefly offer below our answers point by point to the reviewer's comments.
In this work, Vlad and colleagues have developed the study and bioevaluation of a series of 4 O-aryl-carbamoyl-oxymino-fluorene derivatives, exploring their antimicrobial, antioxidant and toxicity attributes.
The authors fully describe the set of results obtained and discuss the major conclusions; however, they present a case for the plasma membrane to be one of the targets of the mode of action, substantiated solely on a membrane depolarization assay. This statement would require some further mechanistic discussion, since the result observed with the DiBAC4 fluorochrome could be the result of non-specific membrane damage, considering the lipophilicity of the compounds under study, as well as a downstream effect of a metabolic inhibition cascade leading to loss of cellular fitness and/or inhibition of energy production. Hence, the reported mechanistic conclusions should be presented as one of many possibilities, or further substantiated.
We thank the reviewer for the observation. Comments have been added in the revised version, in the results and discussion sections.
Furthermore, no relevant discussion was present concerning the solubility limits for the compounds tested, reported as 100 μg /mL for distilled water, and, if applicable, the potential impact in the antimicrobial studies, since the concentrations tested were up to 50 times this concentration, even if in diverse media.
We have prepared the stock solutions in DMSO for increasing the solubility limit of the compounds.
Also, the manuscript could be further improved in some minor points:
- No clear designation of the quality control antibiotics for antimicrobial and anti-biofilm assays.
Considering that the used strains are used as quality control strains in the antimicrobial susceptibility assays and are exhibiting well known profiles to a wide range of antibiotics/antifungals, we did not use a certain antibiotic/antifungal as positive control in our study, as these are available in the CLSI or EUCAST guidelines and currently checked in the laboratory, at least each week.
- Data presented in tables should be standardized in respect to decimal places and/or significant digits
We have standardized the done presented in tables.
Reviewer 3 Report
Considering the current manuscript, this study is intended to evaluate the antimicrobial activities of new O-aryl-carbamoyl-oxymino-fluorene derivatives against different microorganisms. Besides, the antioxidant capacities of these compounds were assessed in relation to ABTS and DPPH. In my point of view, this study introduced many replicated data to expand the data included in this manuscript; for instance, the authors demonstrated the absorbance of ABTS in a table, antioxidant activities in a table, and antioxidant activity in a figure. Similar behaviour was carried out for antimicrobial and antioxidant capacity against DPPH. There are no clear statistical analyses with significant differences in tables and figures. The authors did not elucidate the mechanism of action of these compounds with regard to their antimicrobial and antioxidant properties.
Author Response
We are very grateful to the reviewer for the valuable suggestions. We have carefully addressed them in the revised version of the manuscript and briefly offer below our answers point by point to the reviewer's comments.
Considering the current manuscript, this study is intended to evaluate the antimicrobial activities of new O-aryl-carbamoyl-oxymino-fluorene derivatives against different microorganisms. Besides, the antioxidant capacities of these compounds were assessed in relation to ABTS and DPPH. In my point of view, this study introduced many replicated data to expand the data included in this manuscript; for instance, the authors demonstrated the absorbance of ABTS in a table, antioxidant activities in a table, and antioxidant activity in a figure. Similar behaviour was carried out for antimicrobial and antioxidant capacity against DPPH. There are no clear statistical analyses with significant differences in tables and figures.
We have removed the replicated data, modified the presentation of results for both antioxidant assays (see tables 5 and 7, figures 2 and 3) in manuscript. A clear statistical analysis of the IC50 values for all tested compounds is presented in tables 6 and 8 (see the modified manuscript).
The authors did not elucidate the mechanism of action of these compounds with regard to their antimicrobial and antioxidant properties.
Some of the mechanisms of the antimicrobial action (membrane depolarization) have been highlighted, while the possible mechanism of the antioxidant activity is now discussed in the revised section.
Round 2
Reviewer 3 Report
The revised manuscript is much improved. This work has been thoroughly checked and edited, ensuring that all necessary adjustments have been made. Therefore, I recommend accepting the current version of the manuscript.